# Mixed Reality Enhanced User Interactive Path Planning for Omnidirectional Mobile Robot

**Mulun Wu [1], Shi-Lu Dai [1]** and **Chenguang Yang [2],\***

[1] Key Lab of Autonomous Systems and Networked Control, School of Automation Science and Engineering, South China University of Technology, Guangzhou 510641, China; wumulun@foxmail.com (M.W.); audaisl@scut.edu.cn (S.-L.D.)
[2] Bristol Robotics Laboratory, University of the West of England, Bristol BS16 1QY, UK
\* Correspondence: cyang@ieee.org

**Abstract:** This paper proposes a novel control system for the path planning of an omnidirectional mobile robot based on mixed reality. Most research on mobile robots is carried out in a completely real environment or a completely virtual environment. However, a real environment containing virtual objects has important actual applications. The proposed system can control the movement of the mobile robot in the real environment, as well as the interaction between the mobile robot's motion and virtual objects which can be added to a real environment. First, an interactive interface is presented in the mixed reality device HoloLens. The interface can display the map, path, control command, and other information related to the mobile robot, and it can add virtual objects to the real map to realize a real-time interaction between the mobile robot and the virtual objects. Then, the original path planning algorithm, vector field histogram* (VFH*), is modified in the aspects of the threshold, candidate direction selection, and cost function, to make it more suitable for the scene with virtual objects, reduce the number of calculations required, and improve the security. Experimental results demonstrated that this proposed method can generate the motion path of the mobile robot according to the specific requirements of the operator, and achieve a good obstacle avoidance performance.

**Keywords:** path planning; mixed reality; omnidirectional mobile robot; VFH*

## 1. Introduction

In recent years, with the improvements in production and life automation, the applications of mobile robots are becoming more and more extensive [1–3]. In different scenarios, mobile robots can play different roles, so it is necessary to design and develop robots according to specific user requirements. Therefore, their types and functions are increasingly rich. Efficient path planning is the core technology of a mobile robot, which enables it to move to the designated position in the environments and avoid obstacles safely [4,5]. Different scenarios have different requirements for path planning. Shorter paths can be chosen or safer distances can be considered. In addition, the research and development of much equipment make the control methods of mobile robots more and more abundant, which can meet the interactive tasks of operators in different working environments [6,7].

As a kind of mobile robot, the omnidirectional mobile robot (OMR) has attracted people's attention due to its unique characteristics. It has four Mecanum wheels, each of which has an axis of rotation at a 45-degree angle to the robot body and is driven by a separate motor [8,9]. Therefore, it can carry out lateral and oblique movements that cannot be completed by other constrained mobile robots, due to the advantages of omnidirectional movement and good movement in a limited space [10,11]. The OMR is used in many practical situations, such as the omnidirectional mobile wheelchair [12]. Its motion characteristics make it of high research significance.

For path planning, there are a lot of research results. Current methods include the A* algorithm [13], D* algorithm [14], artificial potential field method [15], vector field histogram* (VFH*) algorithm [16], etc. The VFH* algorithm is a combination of the improved VFH obstacle avoidance algorithm and the A* algorithm. VFH is an object-oriented obstacle avoidance method [17]. Specifically, the vector field histogram is used to represent the surrounding environment of the mobile robot, and the next movement direction is selected according to the cost function of each direction [18]. But this approach sometimes fails to choose the right direction. Therefore, I. Ulrich and J. Borenstein proposed a new VFH* algorithm in [16] to minimize the cost and seek the global optimal solution with heuristic functions. The path planning algorithm used in this paper is an improvement on the original VFH * algorithm. These improvements make the method better applicable to the system proposed in this paper.

Virtual reality (VR) technology simulates a virtual environment by computer to give people a sense of immersion and to allow operators to interact with virtual objects [19–21]. At present, it has been widely used in many fields. Compared with traditional robot control methods, VR can expand the experimental operation space, eliminate security risks in human-robot interaction, and simplify the experimental debugging process [22,23]. Mixed reality (MR) is a technology different from VR, which can superimpose virtual objects into the real environment [24,25]. It also has an important application in robot control [26]. With more and more research on VR and MR, it will certainly bring new developments to many scientific technologies, including robot control.

At present, a lot of research has focused on VR with robot control. In [27], VR control of robot movement is mentioned. The control information is combined with the robot's motion through the Internet, which could be used as a separate simulation system or parallel control with the real motion system. There is also a system for real-time monitoring and networked control of remote robots, which can collect various data about robot operations and complete various assembly tasks [28]. Besides, the robot is controlled by the designed VR technology, and the robot's navigation function is completed in the server, to realize a remote navigation system suitable for simple remote robots [29]. However, the above studies only control the real environment after virtualization, or merely control the virtual robot, and there is no interaction between real robot motion and virtual objects.

This paper proposes a novel path planning interactive system based on MR. With more and more practical applications of VR, mobile robots sometimes need to avoid some virtual objects, such as virtual meetings that need to avoid projecting participants or factories that need to avoid certain locations. The system is proposed for these scenarios. The work of this paper has two main points. Firstly, an MR interaction system was developed to control the mobile robot. Additionally, virtual obstacles can be added to the map of the robot's movement based on specific needs, and path planning can be actively modified. Secondly, the original VFH* algorithm was modified in terms of the threshold setting, the candidate direction selection, and the cost function, making the generated path more suitable for the scene in this paper and more safe and smooth. The remainder of this paper is organized as follows: In Section 2, the system structure of this paper is described. Section 3 introduces the knowledge regarding mixed reality. The dynamics of the omnidirectional mobile robot used in this paper are discussed in Section 4. Section 5 introduces the basic process of the VFH* algorithm and describes the improvement of the algorithm in detail. Section 6 describes the development process of the virtual reality interactive system and the experimental results and effects of path planning. The conclusion of this paper is presented in Section 7.

## 2. System Description

The system structure of this paper is shown in Figure 1; it can control the path planning of the mobile robot through the interface of mixed reality, and realize the interactions between the real environment and virtual objects innovatively. The wearer of the HoloLens uses it to send commands to the computer and receive images. After receiving the commands, the computer combines the corresponding path planning algorithm to generate the optimal path, and then sends it to the mobile

robot to perform the motion process. When the wearer watches the control panel, the wearer can also watch the mobile robot's actual running state, thereby better controlling its movement.

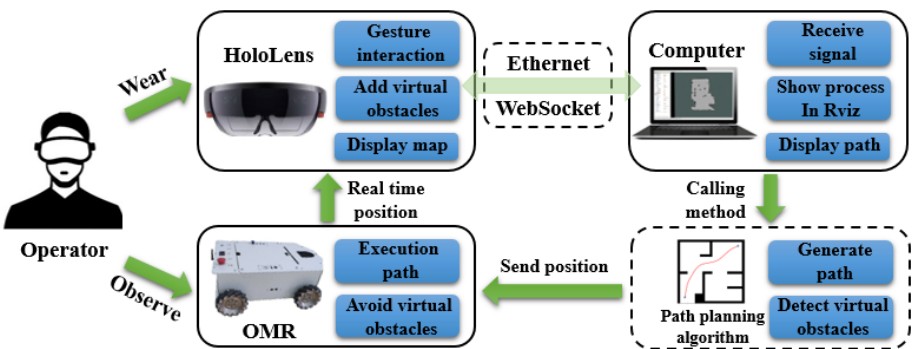

**Figure 1.** Diagram for the system of research.

## 3. Mixed Reality System

### 3.1. Mixed Reality

VR is a familiar technology, and there are many applications in our lives, such as Oculus Rift [30]. In contrast with VR, which completely immerses the user in the virtual environment, MR is a technology that superimposes the virtual environment onto the real environment and allows the user to interact with the virtual environment [31].

In the process of interacting with the robot entity, so as to ensure safety and timeliness, the operator needs to master the state of the robot in time to make corrections or decisions. In robot control based VR, the operator can only operate through the virtual robot transmitted to the device. In practical applications, the delay of transmission and the inaccuracy of the virtual robot state make the effect of this control mode unsatisfactory. But in the robot control combined with MR, the above problems can be well solved. When the operator using an MR device is immersed in the virtual environment, it can also see the real environment. Therefore, the operator can interact with the robot through the virtual environment while looking at the state of the real robot. While giving full play to the advantages of MR, the influence on the control effect of the robot is reduced.

In this paper, MR is used to combine a real scene with virtual objects, and the control system of a mobile robot is designed to avoid virtual obstacles.

### 3.2. Microsoft HoloLens

HoloLens is a head-mounted MR device with a Windows system installed, as shown in the Figure 2. It is equipped with a 32-bit central processing unit (CPU), a graphics processing unit (GPU), and a special holographic processing unit (HPU). The HPU can be used to process a large amount of holographic and sensor data, and achieve motion tracking, spatial positioning, and other functions, which are of great significance in the MR device. Various information fusions of the inertial measurement unit (IMU), the environment understanding camera, a depth camera, and an ordinary camera, enable the device to literally render the virtual environment, giving the wearer a better sense of immersion [32]. In addition, HoloLens has a clear advantage when interacting with robots. As a fully stand-alone device, it can be used independently without wired connection with other hardware during use, which is convenient for the wearer to move with the robot [33]. At the same time, a variety of interaction methods also make the robot control methods diverse.

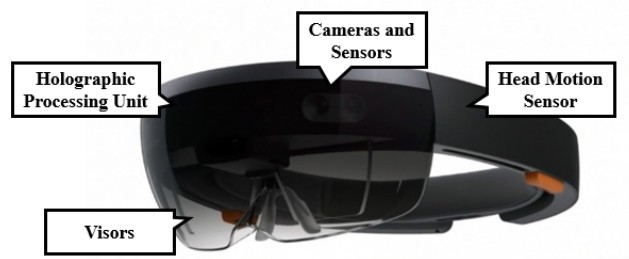

**Figure 2.** The mixed reality device HoloLens.

The development of the HoloLens requires the HoloToolKit development kit, Unity3D, and Visual Studio. The HoloToolKit provides some scripts and components necessary to develop it. Unity3D was used to design a mixed reality application suitable for HoloLens in combination with HoloToolKit, and then Visual Studio deployed the designed application in HoloLens.

In this paper, HoloLens has two functions. One is to send control instructions and position of virtual obstacles to the mobile robot, to control the movement of the robot. The other is to receive the robot motion map and planning path, to master the robot motion information more comprehensively.

## 4. Kinematics of the Omnidirectional Mobile Robot

In this paper, the OMR structure is shown in the Figure 3. It is composed of four Mecanum wheels whose rotational axis is at a 45-degree angle to the robot body, and each pair of diagonal wheels has the same direction [34]. This design allows the mobile robot to move in all directions.

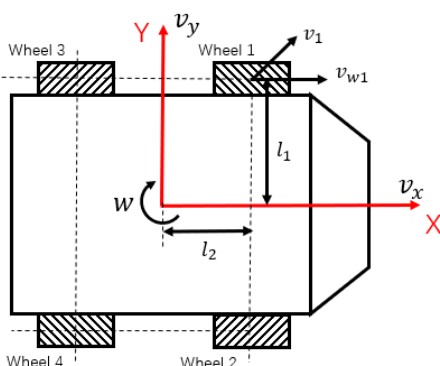

**Figure 3.** Model of the omnidirectional mobile robot.

According to the structure in Figure 3, the velocities of the four wheels of the mobile robot in the x-axis and y-axis can be expressed as follows [35]:

$$v_{x1} = \frac{v_1}{\sqrt{2}} + v_{w1} = v_{rx} + l_1 \cdot \omega \tag{1}$$

$$v_{y1} = \frac{v_1}{\sqrt{2}} = v_{ry} - l_2 \cdot \omega \tag{2}$$

$$v_{x2} = \frac{v_2}{\sqrt{2}} + v_{w2} = v_{rx} - l_1 \cdot \omega \tag{3}$$

$$v_{y2} = -\frac{v_2}{\sqrt{2}} = v_{ry} - l_2 \cdot \omega \tag{4}$$

$$v_{x3} = \frac{v_3}{\sqrt{2}} + v_{w3} = v_{rx} + l_1 \cdot \omega \tag{5}$$

$$v_{y3} = -\frac{v_3}{\sqrt{2}} = v_{ry} + l_2 \cdot \omega \tag{6}$$

$$v_{x4} = \frac{v_4}{\sqrt{2}} + v_{w4} = v_{rx} - l_1 \cdot \omega \tag{7}$$

$$v_{y4} = \frac{v_4}{\sqrt{2}} = v_{ry} + l_2 \cdot \omega \tag{8}$$

where $v_{xn}$ and $v_{yn}$ are the velocities of the four wheels on the x-axis and y-axis, respectively; $v_n$ is the velocity at which the wheels turn; and $v_{wn}$ is the velocity of the wheels relative to the environmental coordinate system ($n = (1, 2, 3, 4)$, represents the number of wheels). $v_{rx}$ and $v_{ry}$ are the overall velocities of the mobile robot; $\omega$ is the angular velocity of the whole robot.

By solving Equations (1)–(8), the relationship between the wheels' velocities with the overall velocity and angular velocity of the mobile robot can be obtained. The relevant formula is as follows:

$$\begin{bmatrix} v_{w1} \\ v_{w2} \\ v_{w3} \\ v_{w4} \end{bmatrix} = \begin{bmatrix} 1 & -1 & l_1 + l_2 \\ 1 & 1 & -(l_1 + l_2) \\ 1 & 1 & l_1 + l_2 \\ 1 & -1 & -(l_1 + l_2) \end{bmatrix} \begin{bmatrix} v_{rx} \\ v_{ry} \\ \omega \end{bmatrix} \tag{9}$$

According to Equation (9), the inverse kinematics equal of the OMR can be expressed as:

$$\begin{bmatrix} v_x \\ v_y \\ \omega \end{bmatrix} = J \cdot \begin{bmatrix} v_{w1} \\ v_{w2} \\ v_{w3} \\ v_{w4} \end{bmatrix} \tag{10}$$

The Jacobian Matrix J is:

$$J = \frac{1}{4} \cdot \begin{bmatrix} 1 & 1 & 1 & 1 \\ -1 & 1 & 1 & -1 \\ \frac{1}{(l_1 + l_2)} & -\frac{1}{(l_1 + l_2)} & \frac{1}{(l_1 + l_2)} & -\frac{1}{(l_1 + l_2)} \end{bmatrix} \tag{11}$$

The full rank Jacobian Matrix J shows that the mobile robot can move in all directions.

## 5. The Description of VFH* Algorithm

In this section, we first introduce the original VFH* algorithm, and then propose the modifications based on it.

### 5.1. The VFH* Algorithm

The VFH* algorithm can be regarded as a path planning algorithm combining the VFH+ local obstacle avoidance algorithm and the A* path planning algorithm. The VFH+ method determines the forward direction of the robot through the following four steps [36]:

(1) Generating the polar histogram: The VFH+ algorithm divides the active region of the current robot position into multiple sectors and calculates the obstacle density in each sector. Then the density of each sector is arranged into histogram according to the sector number.
(2) Binarization polar histogram: One must select the appropriate threshold according to the actual situation, and binarize the histogram generated in the previous step. Sectors above the threshold are set as impassable areas, while sectors below the threshold are set as passable areas.
(3) Mask polar histogram: Considering the kinematics and the dynamics characteristics of the robot, the current inaccessible sectors are set as the impassable areas.

(4)   Determining the direction of motion: The passable areas in the polar histogram are used as the candidate direction; the cost is calculated according to the cost function; and the cost of the passable area is sorted. The commonly used cost function is shown as follows:

$$g(c) = \mu_1 \cdot \Delta(c, k_t) + \mu_2 \cdot \Delta(c, \frac{\theta}{\alpha}) + \mu_3 \cdot \Delta(c, k_p) \qquad (12)$$

where $c$ denote the candidate direction, and $g(c)$ is the cost value of the direction. $\mu_1$, $\mu_2$, and $\mu_3$ are three parameters that we need to determine according to the actual situation. $\Delta(c, k_t)$ is the absolute difference between the candidate direction $c$ and the target direction $k_t$. $\Delta(c, \frac{\theta}{\alpha})$ represents the associated with the difference between the candidate direction $c$ and the robot's orientation $\frac{\theta}{\alpha}$. $\Delta(c, k_p)$ represents the difference between the candidate direction $c$ and the previous direction $k_p$.

Finally, the passable area with the lowest cost is selected as the forward direction of the robot. Since VFH+ only locally plans in real time, global optimization cannot be guaranteed. Therefore, based on VFH+ and A* algorithms, Iwan Ulrich and Johann Borenstein proposed the VFH* algorithm. The A* algorithm is the most effective direct search method for the shortest path in static road network [16]. The VFH* algorithm uses the VFH+ algorithm to predict several future states of the trajectory and form a path tree. Then, the A* algorithm is used to search the path tree, to find the global optimal plan as the next movement direction of the robot's current position.

In this paper, we modified the original VFH* algorithm via the threshold setting, candidate direction selection, and cost function to make it more suitable for the scenario of this paper. When the mobile robot deals with the virtual obstacles suddenly added to the map, it can have a better obstacle avoidance effect.

*5.2. Modification of VFH\* Algorithm*

5.2.1. The Threshold Setting

After generating the polar histogram, the selection of threshold values is very important to determine the direction. If the threshold is too high, the mobile robot will neglect some obstacles on the map, leading to collisions. At the same time, a high threshold will produce more candidate directions and increase the calculation load. If the threshold is too low, the choice of candidate direction will be reduced. When a narrow, passable road section is encountered, the passable direction is neglected by a low threshold, precluding smooth movement. Besides, when virtual obstacles are added into the map, the range of polarity will also increase. Therefore, the threshold of dynamic change shown in the Figure 4 is adopted in this paper. When the mean value of polar coordinates in the polar histogram is less than or equal to the specified value, the small threshold value is selected. When it is greater than the specified value, the large threshold is selected. The change of threshold is related to the mean value of polar coordinates, as shown below:

$$T_1 = \begin{cases} t_{low} & (\sum_n p_n \leq \delta_p) \\ t_{high} & (\sum_n p_n > \delta_p) \end{cases} \qquad (13)$$

where $p_n$ is the polar histogram of each sector and $\delta_p$ is a parameter that distinguishes the size of the density. In the experiment, we used $\delta_p = 1.6$, $t_{low} = 1.2$ and $t_{high} = 2$.

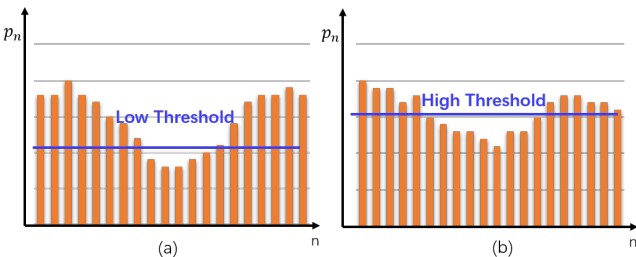

**Figure 4.** The threshold setting policy. (**a**) The low threshold setting. (**b**) The high threshold setting.

There is another special case to consider when selecting a threshold. When the target point is located between the obstacle and the current position of the robot, and the obstacle is within the detectable range. In this case, the direction to the target point is detected as impassable, so the robot cannot reach the target point eventually. To avoid the result, we need to set a special and large threshold for this case to make the direction of the target point passable. Combined with the previous optimization process, the selection of the threshold is as follows:

$$T_2 = max(T_1, \frac{\gamma}{d_{rt}^2})$$ (14)

where $d_{rt}$ is the distance from the mobile robot to the target point, and $\gamma$ is a custom parameter that determines the threshold change condition, in this experiment, $\gamma = 0.5$.

### 5.2.2. The Candidate Direction Selection

After determining the passable sectors, we added the process of sieving the sectors. When the passable sector is the whole circle, which means the robot can move in any direction, the direction towards the target point is undoubtedly a better choice. To avoid unnecessary cost calculation and ensure the accuracy of direction selection, we reduced the passable sector in this case to the area near the target sector, as shown in the Figure 5a.

Besides, when the passable sector is large (greater than 80°), the direction close to the sector boundary is not a good choice. We preferred choosing some directions inside the sector, which were theoretically be safer. Therefore, for the passable sectors, in this case, we also made an appropriate reduction, as shown in the Figure 5b.

As the obstacles in the constructed map will be inflated, the passage area will be safer. However, when some obstacles are close to each other, the operation of obstacle inflation will make the previously passable area impassable due to the narrow detected passable area. To reduce the occurrence of the above situation, when the detected passable sector angle is between 10° to 20°, we will adjust the passable angle to make it passable in the above situation, as shown in the Figure 5c.

If the range of candidate directions does not fall into the above three cases, the generated candidate directions are not adjusted. The above candidate direction selection strategy is expressed as follows:

$$C_d = \begin{cases} [c_{rt} - \dfrac{20°}{\alpha}, c_{rt} + \dfrac{20°}{\alpha}] & (\beta = 360°) \\ [c_l - \dfrac{15°}{\alpha}, c_r + \dfrac{15°}{\alpha}] & (80° < \beta < 360°) \\ \dfrac{c_l + c_r}{2} & (10° < \beta < 20°) \\ [c_l, c_r] & others \end{cases}$$ (15)

where $C_d$ is the range of candidate direction sieved, $c_{rt}$ is the sector from the robot's current position to the target point, $\beta$ is the angle of the passable sectors, $\alpha$ is the angular resolution of sectors, $c_l$ and $c_r$ are the left and right boundaries of the passable sector. We selected $\alpha = 2°$ in this experiment.

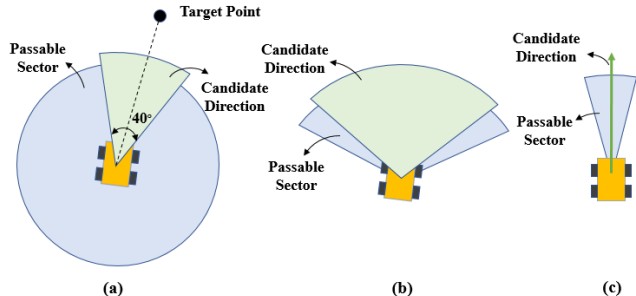

**Figure 5.** The candidate direction selection policy. (**a**) The selection when the passable sector is the whole circle. (**b**) The selection when the passable sector is large. (**c**) The selection when the detected passable sector is small.

### 5.2.3. The Cost Function

After there are several candidate directions, their cost values become the key to which is chosen as the final direction. Taking into account the omnidirectional mobile robot used in this paper—the robot can move in any direction, unlike in Equation (12)—the cost function does not need to consider the current robot angle. Since the VFH* algorithm uses path tree to judge the cost, for a generated path tree, the cost of successor nodes should be related to the cost of previous nodes, which is similar to the following A*algorithm heuristic function:

$$f(n) = g(n) + h(n) \tag{16}$$

where $g(n)$ is the cost from the starting node to the current node $n$, $h(n)$ is the estimated cost from the current node $n$ to the target node, and $f(n)$ is the last cost of the current node $n$. The cost function of VFH* should also be heuristic and include the cost of previous nodes in the path tree. Therefore, we used the following function:

$$\begin{cases} g(c_n) = \lambda_1 \cdot \Delta(c_n, d_t) + \lambda_2 \cdot \Delta(c_n, d_{n-1}) \\ g(c_{n+i}) = \lambda_3 \cdot g(c_{n+i-1}) + \lambda_4 \cdot \Delta(c_{n+i}, d_t) \\ \qquad\quad + \lambda_5 \cdot \Delta(c_{n+i}, d_{n+i-1}) \end{cases} \tag{17}$$

where $c_n$ is the candidate direction from the robot's current position to the next node, $c_{n+i}$ is the candidate direction of the i node in a path tree when generating the tree, $\Delta(c_n, d_t)$ is the absolute difference of the sector between the candidate direction and the target direction, and $\Delta(c_n, d_{n-1})$ is the absolute difference of the sector between the candidate direction and the previous direction.

When calculating the first node with zero-depth in the path tree, the first equation in Equation (17) is used. Only the target node and the previous direction need to be considered. When calculating the successor nodes, the second equation in Equation (17) is used to consider not only the target node and the previous direction, but also the cost of the node before the path tree, and then the cost of this node is synthesized. So, *i* in Equation (17) should be no more than the depth of the path tree.

For parameter selection, the higher $\lambda_1(\lambda_4)$, the more path selection tends toward the target region; the higher $\lambda_2(\lambda_5)$, the smoother the path selection tends to be. To achieve the effect of shortest path, parameters should meet the following conditions:

$$\lambda_1 > \lambda_2, \ \lambda_4 > \lambda_5 \tag{18}$$

At the same time, to make the method more sensitive to sudden virtual obstacles and to avoid obstacles in time to generate a better path, $\lambda_3$ parameters should also meet the following conditions:

$$\lambda_3 \cdot \lambda_1 < \lambda_4, \ \lambda_3 \cdot \lambda_2 < \lambda_5 \tag{19}$$

After the path tree of a node is generated, the mobile robot will move the distance of a node in the direction with the lowest total cost in the tree, and then generate a new path tree at the location of the new node. Repeat the above steps until the robot reaches the target point.

In the experiment, the relevant parameters of the cost function were as follows:

$$\lambda_1 = 5, \lambda_2 = 3$$
$$\lambda_3 = 0.5, \lambda_4 = 3, \lambda_5 = 2$$

(20)

Combined with the original process of VFH* algorithm and the modification proposed in this paper, the implementation process of the system obstacle avoidance algorithm is shown in Algorithm 1:

---
**Algorithm 1** The improved VFH* algorithm

---
**Input:** Map information, start point and target point location;
**Output:** Trajectory;
  1: **while** the distance between the robot and the target point $\leq 0.1$ **do**

  2:     Generate polar histogram from current position;
  3:     The passable candidate direction is screened by combining threshold $T_2$ and mask $C_d$;
  4:     The heuristic function $g(c_n)$ and $g(c_{n+i})$ are used to calculate the cost of passable reference

        direction;
  5:     Choose the direction with the least cost as the final direction;
  6:     Transmit the direction;
  7: **end while**;

---

## 6. Experimental Results and Analysis

### 6.1. The Design of the Interactive System

The interactive system designed in this paper included the interactive interface of HoloLens, the control of the mobile robot, and the communication between the two parts.

The interface was a virtual panel and some buttons with Unity3D software combined with HoloToolKit, as shown in Figure 6. The panels can show the map and the mobile robot's trajectory, and the buttons can interact with the movement of the mobile robot. These buttons in the interface can be used to establish communication with the mobile robot and receive the map and robot position, control the start and stop of the robot, and set the position of the target point and obstacles through gesture interaction. This control mode gives full play to the advantages of HoloLens, which can not only see the real motion scene of the robot through the lens but also operate the projected virtual panel only through gestures without the help of other devices.

We then built the project in a Visual Studio solution file in Unity3D, and used Visual Studio to deploy the project to HoloLens via USB cable. When HoloLens and the mobile robot are on the same LAN and their IP addresses are matched, HoloLens can control the mobile robot independently.

The communication mode in this system adopted the flow shown in Figure 7. Communication between the interface in HoloLens and robot operating system (ROS) was conducted through ROSBridge WebSocket. After receiving the information, the ROS system would send it to the ROS plugin related to the path planning algorithm and map construction. Finally, the calculated motion control information would be sent to the mobile robot that subscribed to it through the ROS topic. HoloLens received information from ROS in the same way. Compared with other communication protocols, WebSocket can frequently carry out two-way data transmission and maintain a persistent connection between communication parties, which is more suitable for real-time control between robot and HoloLens during robot movement in this system.

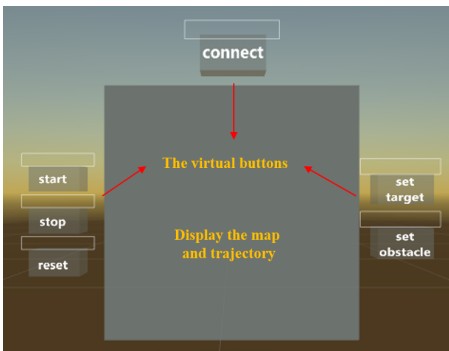

**Figure 6.** The virtual panel in Unity3D.

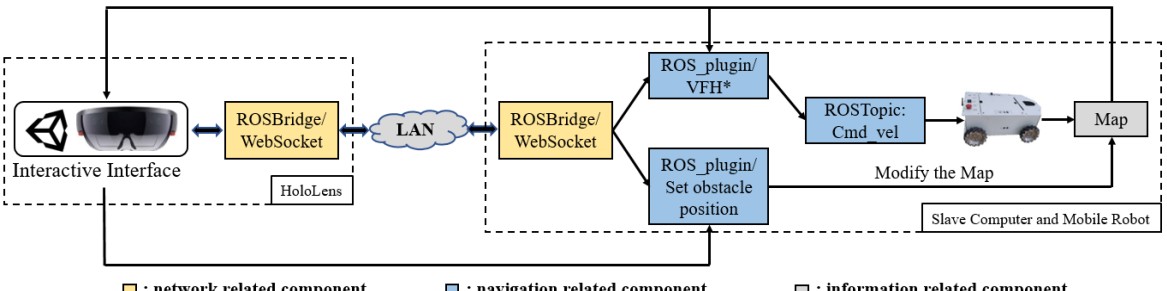

**Figure 7.** Diagram of software and communication layout.

The complete system operation process is as follows:

(1) Establish a connection to synchronize the pre-acquired map and the initial coordinates of the mobile robot;
(2) Set the moving target point and start the robot;
(3) In the process of robot movement, choose whether to add virtual obstacles and their positions;
(4) Transmit the current position of the real robot to the virtual panel in real-time until it reaches the target point.

Because of the network transmission delay, the position of the robot in the virtual panel will be slightly delayed from its actual position, but the operator can see the robot's movement in the real scene at the same time, so the impact of this small delay is not great.

*6.2. Path Planning without Virtual Obstacles*

This part of the experiment tested the actual effect of the proposed path planning algorithm on the known map without virtual obstacles added. The map adopted was a 16 × 28 m simulation environment, in which we put some fixed obstacles to test the effect of obstacle avoidance. One starting point and two target points were set, which made the planning path longer and the experimental results more universal. In order to better show the experimental effect, we used a dynamic window approach (DWA) in the ROS navigation package as the comparison algorithm under the condition that other variables were the same.

Based on two different path planning algorithms, the mobile robot started from the same point on the map and passed through two target points, respectively. The trajectory generated on RViz was shown in the Figure 8. The comparison of trajectory coordinates is shown in the Figure 9a. As can be seen from the figures, both algorithms can enable the mobile robot to successfully avoid obstacles and reach the target point. But the improved VFH* produces a smoother trajectory, which makes the movement smoother. On the other hand, the trajectory generated by the proposed algorithm distance from the obstacle was safer. When a narrow passage was encountered, the middle safer trajectory was chosen.

The comparison of the changes in velocity and angular velocity of the two algorithms in the process of motion is shown in Figure 9b,c. When adopting the algorithm proposed in this paper, the maximum and minimum of velocity and angular velocity respectively, were limited, and the deceleration when obstacles were detected was optimized, so that it would not be reduced to zero directly. In this way, the mechanical motion structure of the mobile robot can be better protected and the moving process can be smoother. Meanwhile, by comparing the two figures, it can be seen that the two curves suddenly changed at the same time: when the mobile robot detected obstacles to avoid.

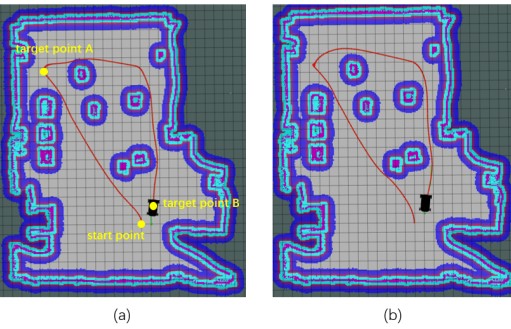

(a)                                    (b)

**Figure 8.** Trajectory comparison in RViz. (**a**) Trajectory using the DWA algorithm. (**b**) Trajectory using the improved VFH* algorithm.

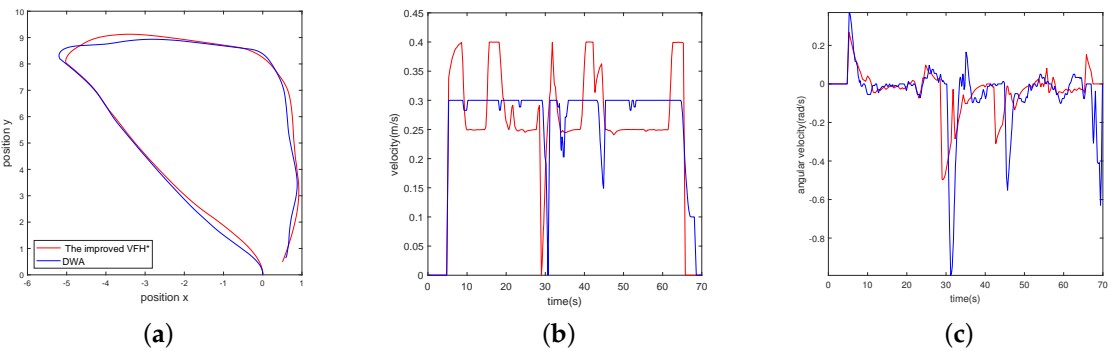

(**a**)                          (**b**)                          (**c**)

**Figure 9.** The results of different path planning methods were compared in the experiment I. (**a**) The trajectory comparison. (**b**) The velocity comparison. (**c**) The angular velocity comparison.

Additionally, as can be seen from Table 1, under the premise of setting the same starting point and target points, the improved method in this paper produces shorter path lengths, shorter robot movement time than the traditional DWA method, and the sum of the distances to the obstacles is further when the robot passes the same number of obstacles.

**Table 1.** Experimental results of parameters in the path planning without virtual obstacles.

| Methods | DWA | The Improved VFH* |
|---|---|---|
| Path length (m) | 17.91 | 17.36 |
| Run time (s) | 64.6 | 62.2 |
| Passing obstacle | 5 | 5 |
| Sum of the distance from the obstacle (m) | 2.95 | 3.40 |

The improved algorithm proposed in this paper was effective in this experiment. The mobile robot could smoothly avoid obstacles and safely reach the specified target point. Besides, the changes of velocity and angular velocity were flexible and the system was robust.

### 6.3. Path Planning with Virtual Obstacles

This part of the experiment showed the path planning effect of the mobile robot after adding virtual obstacles to the map with HoloLens. We used the same map as the previous experiment and set a starting point and a target point. The experimental operation process was as follows: When the mobile robot moved to a certain position, two virtual obstacles were added to the map through gesture control of HoloLens. Then, the path planning algorithm took these two virtual obstacles into account and re-planed the path. The scenes in HoloLens before and after the operation of adding obstacles are shown in Figure 10a,b. The actual path to avoid virtual obstacles in RViz is shown in Figure 10c,d. The trajectory comparison of before and after the addition of virtual obstacles is shown in Figure 11a. It can be seen that in the process of motion, when the mobile robot detected the suddenly added obstacle, it would immediately change the previously planned trajectory, and it then re-planed the trajectory according to the new map and moved according to the new trajectory.

The changes of velocity and angular velocity are shown in Figure 11b,c. The limitation of the velocity was also to protect the mobile robot. After adding obstacles, the two curves would change obviously due to the sudden detection of obstacles. When adding virtual obstacles, its position could not be too close to the moving mobile robot, otherwise, obstacle avoidance would be confused.

Figure 12a,b respectively show the performances of the traditional VFH* path planning algorithm and the DWA algorithm in the scene where virtual obstacles were suddenly added. It can be seen that neither of the two methods could complete the experiment well. When using the DWA method, it was difficult to update the path in time, so after adding virtual obstacles to the map, the mobile robot did not correct the path to avoid the obstacles, but still moved according to the original path, failing to reach the target point. Compared with the traditional VFH* method, the main reason why the improved VFH* has a better effect is that the new cost function fully considers the characteristics and requirements of the system. Since the platform used was OMR, the unaffected part of the original function was removed, and the coefficient of the function was modified to make the algorithm more sensitive to the virtual obstacles suddenly added in the distance and to adjust the path in time.

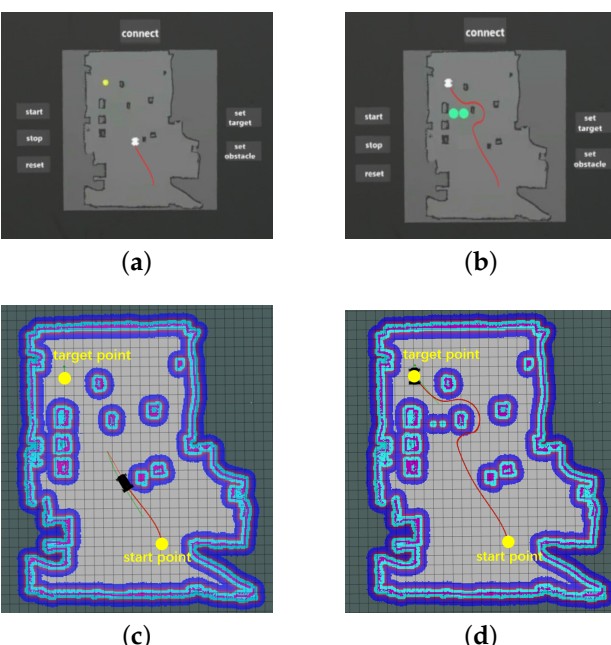

(a)  (b)

(c)  (d)

**Figure 10.** The trajectory in HoloLens and RViz before and after adding virtual obstacles. (**a**) The scene in HoloLens before adding virtual obstacles. (**b**) The scene in HoloLens after adding virtual obstacles. (**c**) The scene in RViz before adding virtual obstacles. (**d**) The scene in RViz after adding virtual obstacles.

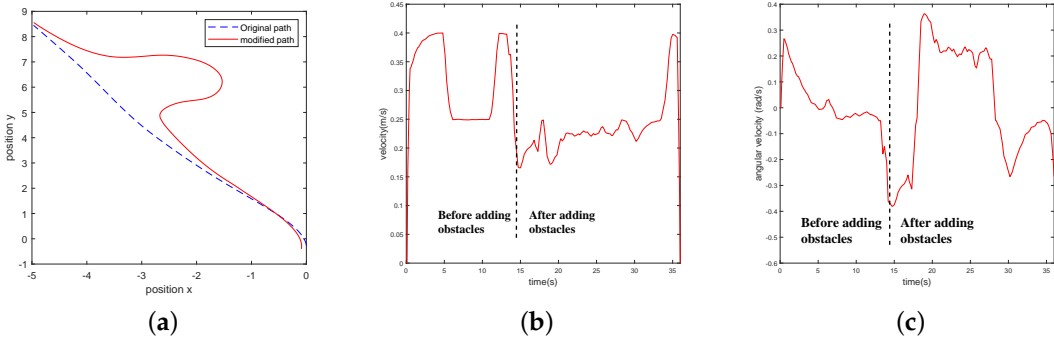

**Figure 11.** The results before and after adding virtual obstacles in experiment II. (**a**) Trajectory comparison before and after adding obstacles. (**b**) Change in velocity before and after adding obstacles. (**c**) Change in angular velocity before and after adding obstacles.

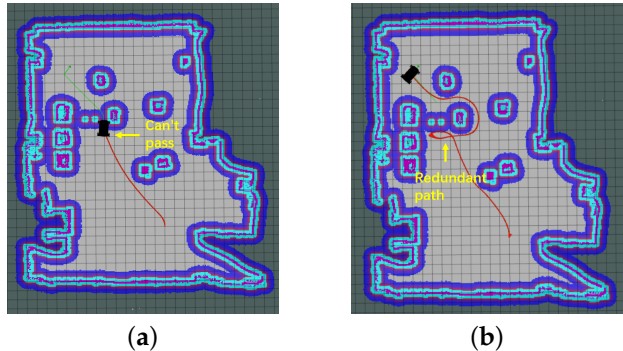

**Figure 12.** Comparisons with the experimental results of other path planning methods. (**a**) The path planning effect using the DWA method. (**b**) The path planning effect using the traditional VFH* method.

### 6.4. Real Experimental Validation

To verify the actual effect of the system proposed in this paper in real situations, we tested the system in a real experimental environment using the mobile robot mentioned in this paper. The experimental environment was limited in a size of 3.5 × 3 m, and three fixed obstacles were placed. This part of the experiment tested a variety of scenes without or with virtual obstacles. The movement process of the mobile robot in its experimental environment is shown in Figure 13.

Consistent with the simulation results, the system can directly control the mobile robot for path planning, and set up virtual obstacles by a virtual panel to make the mobile robot avoid the location we do not want to pass and reach the target point successfully. The system can be well applied to such situations as virtual meetings that need to avoid some unreal obstacles.

To better illustrate the effect of this applied research, we evaluated the operator's experience from the perspectives of operating equipment number, operating time, and convenience. When using the system designed in the paper, operators can send commands simply by interacting with a device called HoloLens through gestures. When using a computer for control, it also needs to be controlled by peripherals such as keyboard and mouse, so the control end needs at least three devices, including the host. In terms of operation time, because the system integrates all the instructions, they can be implemented on the same panel. The total operation time for the above four physical experiments was 13 s, while when directly controlled by computer, different instructions were input on different interfaces, and the total time was 27 s. Finally, when the HoloLens is used for control, the control panel and the actual scene are in the same perspective, which is more convenient for control and can have a better interactive effect with the virtual object. However, when the computer is used for control, the operator needs to observe the screen and the actual scene at the same time, and there is no

immersion effect. Therefore, compared with the traditional method, using this system to control the mobile robot is more efficient and more convenient.

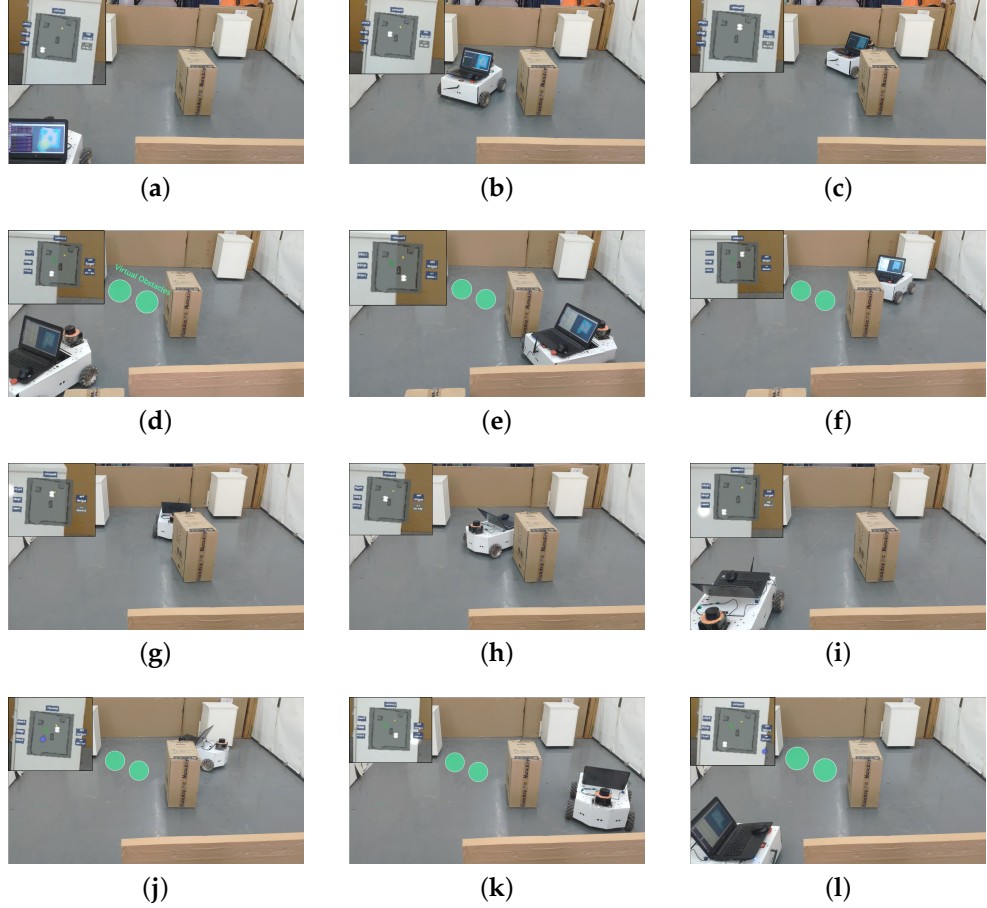

**Figure 13.** Verifying the results of the experiment with the real omnidirectional mobile robot. The top left corner of each picture is the image in HoloLens. Since HoloLens superimposes virtual objects on a real scene, the cabinet in the real scene can be seen in the image. (**a**–**f**) The process comparisons of reaching the target point without and with virtual obstacles, respectively. The green circles are the virtual obstacles that do not exist in the real scene and only act on the cost map of the mobile robot. (**g**–**l**) Comparisons of processes that return to the starting point without and with virtual obstacles, respectively.

## 7. Conclusions

In this paper, a novel mobile robot path planning interactive system combined with MR is proposed to meet the needs of real robots and virtual-object-interaction scenes. An MR interaction system was designed, in which the MR device HoloLens can either simply control the mobile robot to do the path planning in the real map, or combine virtual objects to plan the path to meet specific requirements. At the same time, the original VFH* algorithm can be improved in terms of the threshold setting, the candidate direction selection, and the cost function, so as to generate a safer and smoother path, which is suitable for the real scenario in this paper. The path planning in the paper can also change other methods according to the requirements of practical applications, but the methods need to be optimized according to the characteristics of the interactive system. Using MR to control the mobile robot's movement has many advantages. MR allows the operator to see the real environment of the mobile robot while seeing the virtual objects, making the control more secure and real-time. At the same time, the interaction between virtual objects and the real environment can meet the needs of the increasing number of virtual environments. The integration of HoloLens enables the operator

to realize the remote interaction of the mobile robot with less equipment. In future work, we will continue to focus on the research and applications of VR and MR in the field of robot control.

**Author Contributions:** M.W. conceived the method, performed the experiments and wrote the paper; C.Y. conceived the method and helped to improve it; S.-L.D. further helped to improve it. All authors have read and approved the manuscript.

**Funding:** This work was partially supported by Foshan Science and Technology Innovation Team Special Project under Grant 2018IT100322 and National Nature Science Foundation (NSFC) under Grant 61973129.

**Conflicts of Interest:** The authors declare no conflict of interest.

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
