# Peer review of "Mixed Reality Enhanced User Interactive Path Planning for Omnidirectional Mobile Robot"

_applsci, doi:10.3390/app10031135_

Round 1
Reviewer 1 Report
This work proposes a novel control system for path planning of an omnidirectional mobile robot based on mixed reality. In this paper, an interactive system based on mixed reality is designed and the method of path planning is improved for the system. Generally, the combination of mixed reality and path planning is an innovative attempt, and the theoretical part and experimental part are relatively complete, can better explain the effect of the system. My comments are listed below:
(1) HoloLens is just a device and platform, and using it as a keyword is not a good choice.
(2) In formula (13), The selection condition of the formula should be more accurate by adding an equal sign.
(3) In the section of “4.2.2 The candidate direction selection”, the selection condition of candidate direction does not include all the angles. So what happens when you encounter an Angle that's not in the formula? Please explain the formula more to make it easier to understand.
(4) In Figure 5 (c), the first "the" should be capitalized. Except for this, I found many typos throughout the paper. The authors have to proofread their paper carefully.
(5) The accurate description in Figure 10 should consist of two parts: one is the correspondence between the running processes of HoloLens and RViz, and the other is a comparison with the different path planning methods. Combining the two parts into one figure will confuse the reader, so it is recommended to divide it into two diagrams.
(6) In the section of “5.3 Path Planning with Virtual Obstacles”, the improved algorithm is significantly different from the original one. Can you give a more detailed explanation based on the previous theoretical part?
(7) The information about some references is also inaccurate. Please double check.
Reviewer 2 Report
The paper presents a virtual reality application for supporting an interactive path planning approach. In particular, the authors propose a mixed reality system to allow virtual obstacle definition and a VFH* variant to better address obstacle avoidance.
The whole architecture may be better presented. Currently, the architecture is presented in the Introduction while a dedicated section would facilitate the readability.
The work seems interesting and well motivated. Even though not groundbreaking, there are improvements from the proposed variant of VFH*.
The experimental validation shows positive results both under simulation that in realistic scenarios.
It is not fully clear to me the actual impact of the virtual reality interface. Authors should better emphasize which are the most significant advantages in pursuing such kind of approach. Also, in addition to a performance evaluation, it would be interesting to have also a user experience evaluation.
In general, the paper describes a nice work.
Reviewer 3 Report
The paper presents a control system for path planning of an OMR using a modified version of the VFH algorithm and MR.
The article is generally well written. Some modifications are proposed below:
A more detailed description of the algorithm would be helpful. Using pseudo-code and diagrams would be very useful.
A clearer presentation of the results obtained and a comparison with the state of the art would be recommendable.
Certain paragraphs in the document seem slightly "badly linked".
